# Census Tract Food Tweets and Chronic Disease Outcomes in the U.S., 2015–2018

**DOI:** 10.3390/ijerph16060975

**Published:** 2019-03-18

**Authors:** Yuru Huang, Dina Huang, Quynh C. Nguyen

**Affiliations:** Department of Epidemiology and Biostatistics, University of Maryland School of Public Health, College Park, MD 20742, USA; yorohuang@gmail.com (Y.H.); dhuang12@umd.edu (D.H.)

**Keywords:** Twitter, food environment, chronic disease

## Abstract

There is a growing recognition of social media data as being useful for understanding local area patterns. In this study, we sought to utilize geotagged tweets—specifically, the frequency and type of food mentions—to understand the neighborhood food environment and the social modeling of food behavior. Additionally, we examined associations between aggregated food-related tweet characteristics and prevalent chronic health outcomes at the census tract level. We used a Twitter streaming application programming interface (API) to continuously collect ~1% random sample of public tweets in the United States. A total of 4,785,104 geotagged food tweets from 71,844 census tracts were collected from April 2015 to May 2018. We obtained census tract chronic disease outcomes from the CDC 500 Cities Project. We investigated associations between Twitter-derived food variables and chronic outcomes (obesity, diabetes and high blood pressure) using the median regression. Census tracts with higher average calories per tweet, less frequent healthy food mentions, and a higher percentage of food tweets about fast food had higher obesity and hypertension prevalence. Twitter-derived food variables were not predictive of diabetes prevalence. Food-related tweets can be leveraged to help characterize the neighborhood social and food environment, which in turn are linked with community levels of obesity and hypertension.

## 1. Introduction

The food environment, including access to grocery stores and different types of restaurants, may shape an individual’s food choices as well as eating habits, which could influence chronic disease outcomes [1]. Researchers have extensively studied the neighborhood food environment in relation to obesity. Food environment indicators, such as a short distance from supermarkets, more favorable retail food environments [2], and less access to fast food restaurants [3], have all shown to be associated with lower obesity rates [4]. Similarly, the prevalence of diabetes and hypertension are affected by local food environments [5,6].

Social media, such as Twitter and Facebook, provides a massive amount of user-generated data. Public health researchers have used social media data to track the spread of communicable disease outbreaks. For example, Wakamiya et al. used Twitter data for influenza detection [7] and Signorini et al. leveraged Twitter data to track disease activities and public concerns [8]. There is, however, increasing use of social media data for neighborhood environment characterization [9]. Twitter data have been used to show the geographical variation in diet choices and nutrition, which furthers the understanding of food environments in different neighborhoods and identifies “food deserts” in certain regions [10]. The information retrieved from Twitter is a reflection of the Twitter user’s health behavior, as well as their utilization of neighborhood resources [11].

Social learning theory [12] can be used to explain the reciprocal and dynamic interactions of the person, online Twitter environment, and diet choices. The theory posits that individuals have the ability to adopt healthy behaviors by observing the modeling of behaviors. Users would tweet about what they eat [13], which may encourage certain eating styles in their local social network. Social norms about healthy eating (i.e., socially accepted behaviors) can be communicated on Twitter and shape an individual’s food choices [14].

Most previous research that examined the relationship between neighborhood food environments and cardiometabolic outcomes utilized health surveys. However, neighborhood surveys can be costly and become quickly outdated. The food environment indicators that were previously used, such as distance from supermarkets, retail food environments [2], and access to fast food restaurants [3], do not capture individuals’ food choices and their interactions with the local food environment. Social media data offer a novel resource for neighborhood research. Dynamic and real-time Twitter data are directly reflective of user activity, which provide more in-depth information on individual health behaviors across the nation.

In this study, we sought to investigate the relationship between aggregated food-related tweet characteristics at the census tract level and the prevalence of certain cardiometabolic outcomes such as obesity, diabetes, and high blood pressure, utilizing geotagged Twitter data collected from 2015 to 2018. We obtained a prevalence of cardiometabolic outcomes at the census tract level from the CDC 500 Cities project [15], which collected small area estimates for chronic health risk factors, health outcomes and health services for the 500 largest cities in the U.S. Our study advances understanding of the relationship between the food environment and cardiometabolic outcomes at a small area level across the U.S., providing information for public health interventions.

## 2. Materials and Methods

We utilized the Twitter streaming application programming interface (API) to collect 1% of random tweets from April 2015 to May 2018. Only tweets with a pair of coordinates of where they were sent (geotagged) were collected. A total of 4,785,104 geotagged food tweets were obtained from 71,844 census tracts across the contiguous United States. Tweets related to job postings were excluded. The tweets were then filtered by a list of food keywords (n = 1430) created from the U.S. Department of Agriculture’s National Nutrient Database [16]. Each food item in the list was linked to a measure of caloric density (calories per 100 g). Fruits, vegetables, nuts and lean protein were classified as healthy food items (340 keyword terms). Additionally, 154 keyword terms were used to capture the mention of fast food restaurants. We chose fast food restaurants as an indicator of worse food environment because previous research has found that people who tweeted about fast food had more fast food restaurants in their surroundings than green retailers [11]. Living farther away from a fast food restaurant was found to be associated with lower body mass index (BMI) for children [17].

Only geotagged tweets related to food were retained to construct indicators of the neighborhood food environment. Each tweet was searched for food items in the food keyword list and was labeled “1” for healthy food if a healthy food item was found in the tweet. Similarly, a tweet was labeled “1” for fast food, if a fast food item was found. Caloric density was calculated by summing up the caloric density for all the food items mentioned in a single tweet. Detailed methods of Twitter data processing were described in a prior paper [9]. By summarizing all the tweets within a given census tract level, we created the following census tract food environment variables: Average caloric density, the percentage of food tweets that mentioned a healthy food, and percentage of food tweets that mentioned fast food. The Twitter-derived food environment characteristics were then standardized to ease the interpretation of results. We have made our census tract level data publicly available at the following location: https://hashtaghealth.github.io/.

We obtained census tract level health outcomes, including the prevalence of obesity, diabetes, and hypertension, from the 500 Cities project from the Centers for Disease Control and Prevention (CDC). The 500 Cities project is a collaborative project launched by the Robert Wood Johnson Foundation, CDC Foundation and CDC, in 2015 [15]. The project aims to provide small area estimates of chronic health outcomes including city level and census tract level estimates for the 500 largest cities across the U.S. The project included 497 of the largest cities and additionally added 3 cities in Vermont, West Virginia and Wyoming to guarantee the inclusion of all the states (including Alaska and Hawaii) in the U.S. [15]. According to the U.S. Census 2010, these 500 cities represent 33.4% of the total population from approximately 28,000 census tracts [15].

Census tract level demographics were obtained from the American Community Survey (ACS) 2014 five year estimates and included the population density, percent 65 or older, percent male, percent black, percent Hispanic, urban or rural areas, percent relatives (besides spouse and children) living in households, percent unmarried cohabitating adults, household size, percent owner-occupied housing, and income inequality.

To ensure the construct validity of food environment characteristics, we excluded census tracts that had less than 10 tweets collected (n = 4820). A total of 23,324 census tracts were included after merging 500 Cities’ health outcomes to the Twitter-derived food environment for the contiguous United States. We used quantile regression to estimate associations between the Twitter-derived neighborhood characteristics and the conditional median prevalence of health outcomes, which is more robust to outliers. Each outcome (obesity, diabetes and hypertension) was modeled separately. The models controlled for census tract demographics. An adjusted linear regression was performed as a sensitivity analysis for each of the outcomes. We additionally performed a generalized linear model using generalized estimating equations (GEE) to assess temporal trends in average calories per tweet, by state. Moreover, to assess the impact of the Twitter-derived food environment on chronic outcomes in subgroups, we obtained county level sex-specific obesity rates from the CDC 2013 Behavioral Risk Factors and Surveillance System (BRFSS), a telephone survey on the health behaviors related to chronic diseases, injury, and preventable infectious diseases, for the non-institutionalized, adult U.S. population. Age-adjusted estimates for percent obese were generated from these data using a small area estimation technique [18,19]. Separate models were fit by sex to examine associations between the Twitter-derived food environment characteristics and sex-specific obesity rates. R was used for all the statistical analysis [20].

## 3. Results

The percent of food tweets mentioning fast food was highest in Idaho (23.33%), Kentucky (21.57%), West Virginia (19.53%) and Michigan (18.94%) (Figure 1). The average caloric density was highest in West Virginia (270 calories per tweet) (Figure 2). The average caloric density was lowest in Montana (154 calories per tweet) and the fast food mentions for Montana (8.09%) was among the lowest. The percent of food tweets mentioning healthy food was the highest in Mississippi (20.19%), Oregon (19.57%) and Ohio (17.67%) (Figure 3). Oregon (6.80%) and Mississippi (10.47%) also had fewer fast food mentions. Detailed food environment characteristics by state can be found in Appendix A.

A one standard deviation increase in caloric density was associated with a 0.19% increase in the prevalence of obesity and a 0.16% increase in the prevalence of hypertension (Table 1). We observed similar results for fast food mentions, where a one standard deviation increase was associated with a 0.15% increase in the prevalence of obesity and a 0.12% increase in the prevalence of hypertension. The percent healthy food was inversely associated with the prevalence of obesity and hypertension. A one standard deviation increase in percent healthy food was associated with a 0.30% decrease in the prevalence of obesity and a 0.13% decrease in the prevalence of hypertension. In sensitivity analyses, we fit adjusted linear regression models for each outcome separately (Appendix A). Similar to the results from the median regression, average caloric density and fast food mentions were associated with a higher prevalence of obesity and hypertension. Healthy food was associated with a lower prevalence of obesity and hypertension. None of the Twitter-derived food environment characteristics were associated with the prevalence of diabetes.

In additional analyses, we examined temporal trends using GEE models. The average calories per food tweet increased 15.87 kcal each year (*p* < 0.001), the percent of fast food mentions increased 1.37% each year (*p* < 0.001), and the percent of healthy food mentions increased 0.59% (*p* < 0.001) each year. Figure 4 displays temporal trends for the average calories by state. The x-axis displays year (2015–2018) and the y-axis displays average calories. The majority of states had an upward trend for average calories between 2015 and 2018.

In analyses with sex-specific obesity rates (Appendix A), a one standard deviation increase in calories was associated with a 0.32% increase in the prevalence of obesity in males and a 0.42% increase in the prevalence of obesity in females. Similarly, a one standard deviation increase in the percent fast food mentions was associated with a 0.13% increase in the prevalence of obesity in males and a 0.21% increase in the prevalence of obesity in females. Nonetheless, when we conducted a Chow test to assess the differences between the separate regression models, we found that there was no statistically significant difference between the coefficient estimates for the Twitter-derived variables in the male-specific obesity model versus those in the female-specific obesity model, suggesting a similar strength of associations between the Twitter food environment variables and the sex-specific obesity rates.

## 4. Discussion

We found that Twitter-derived food environment characteristics are predictive of obesity and hypertension prevalence at the census tract level in the United States. A higher caloric density, higher percentage of fast food mentions, and lower percentage of healthy food mentions, among all food tweets in the neighborhood, were positively associated with a higher prevalence of obesity and hypertension. Twitter-derived food characteristics were not associated with diabetes.

Findings from our work point to a relationship between the neighborhood food environment and diet-related health outcomes, which has been supported by previous research using traditional research methods [21,22,23]. The underlying mechanism is well studied between fast food consumption, increased caloric intake and obesity risk [24]. The neighborhood food environment may influence fast food consumption and therefore change the obesity risk for residents. Hypertension and diabetes are also heavily influenced by diet choices [5,25].

The relationships between food mentions and census tract level health outcomes identified from our study were generally consistent with prior neighborhood research findings utilizing Twitter data. Previously, Abbar et al. used social media data to monitor dietary habits and found that the food mentions in the daily tweets of users were predictive of obesity and diabetes statistics at the state level [13]. Similar studies leveraging other social media data sources, such as Facebook, also demonstrated the relationship between online social environment characteristics and the prevalence of obesity at the city level using data from the Behavioral Risk Factor Surveillance System and the Selected Metropolitan/Micropolitan Area Risk Trends (SMART) project, and these relationships were also seen at the zip code level in New York City [26]. Previously, Twitter food environment characteristics (tweets collected from 2015 to 2016) were demonstrated to relate to state level health outcomes. Higher caloric density of food mentions was associated with higher prevalence of diabetes [27].

This is the first study that analyzed Twitter-derived food environment characteristics and its relationship to chronic health outcomes at the census tract level, which is able to incorporate heterogeneity in food tweets characteristics within each state. Food environments within each state can be vastly different—from small disadvantaged areas to wealthy suburbs—therefore, census tract level analyses may be more able to reflect an individual’s local food environment. Collecting Twitter data over a span of three years also provided us with robust data to construct national neighborhood indicators. We utilized health outcome data obtained from the CDC 500 Cities project. Hence, health outcome data also came from geographically diverse areas. To explore the impact of the Twitter-derived food environment characteristics on chronic outcomes by subgroups, we additionally performed analyses for sex-specific obesity rates. We found that there were not statistically significant differences in the strength of associations between the Twitter-derived variables and county obesity rate for men vs. women.

This study is subject to several limitations: The 500 Cities project includes data from the most populous cities in the United States and covers about 33.4% population. Therefore, our results may not be representative of the U.S. population and may not generalize to less populous cities or rural neighborhoods. Further studies are needed to examine the relationship between Twitter characteristics and chronic health outcomes in smaller cities or rural areas. We utilized 1% of publicly available tweets with geotags collected from Twitter’s free API service. It was estimated that about 28% of the online users possessed a Twitter account in 2018 [28] and Twitter users tend to be younger in age [29]. Therefore, the information collected from Twitter may not be representative of all views expressed online or in-person at the neighborhood level. Facebook remains the most popular social media in the U.S. [29]. However, the majority of profiles on Facebook are private, whereas, the majority of profiles on Twitter are public and thus available for analyses. Moreover, it is important to bear in mind that only a small number of tweets are geotagged and may not fully represent the food environment of a given area. However, although tweets may not be representative of all people, behaviors or food resources, as this study demonstrates, tweets may still have utility in predicting health outcomes. Activity on social media itself may impact health behaviors and outcomes. We did not have information on internet use or social media use. Our analyses did account for rurality, minority composition, and socioeconomic status, which correlate with internet access and use [30]. In addition, we were not able to perform a longitudinal analysis, due to the lack of health outcome data from 2016 to 2018. The cross-sectional study design had its inherent limitations. This study design could not definitely determine if a neighborhood food environment caused more obesity and hypertension cases; we report on associations observed. Our study also used aggregated data, therefore inferences cannot be drawn at the individual level.

## 5. Conclusions

We find that Twitter-derived food environment characteristics including higher average calories per tweet, less frequent healthy food mentions, and a higher percentage of food tweets about fast food are linked with higher obesity and hypertension prevalence at the census tract level. Harnessing information exchanged online can be a cost-effective way of understanding local health trends and providing information necessary for targeted interventions. Neighborhood Twitter studies can help to identify areas at risk, inform chronic disease prevention at a neighborhood level, and further contribute to reducing important and costly chronic diseases. Our Twitter-derived neighborhood data are publicly available at https://hashtaghealth.github.io/.

## Figures and Tables

**Figure 1 ijerph-16-00975-f001:**
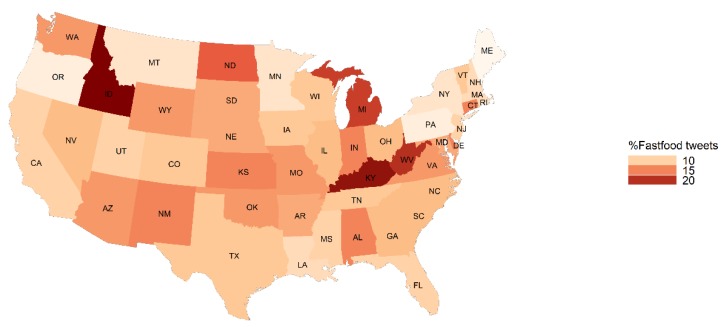
State level percent of food tweets with a fast food mention in the contiguous United States, Twitter 2015–2018.

**Figure 2 ijerph-16-00975-f002:**
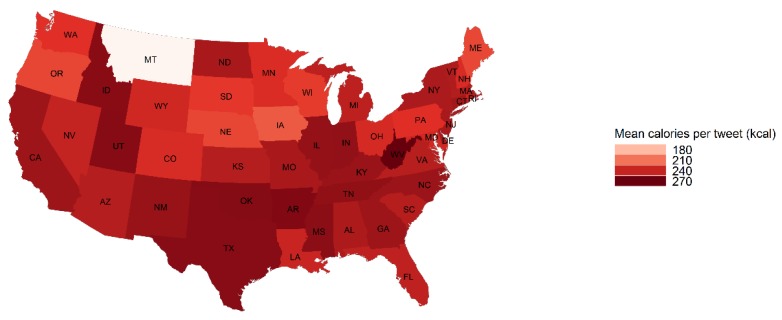
State level average caloric density per food tweet in the contiguous United States, Twitter 2015–2018.

**Figure 3 ijerph-16-00975-f003:**
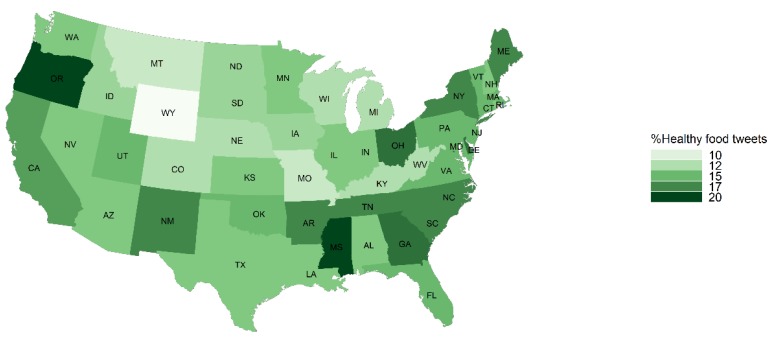
State level percent of food tweets with a healthy food mention in the contiguous United States, Twitter 2015–2018.

**Figure 4 ijerph-16-00975-f004:**
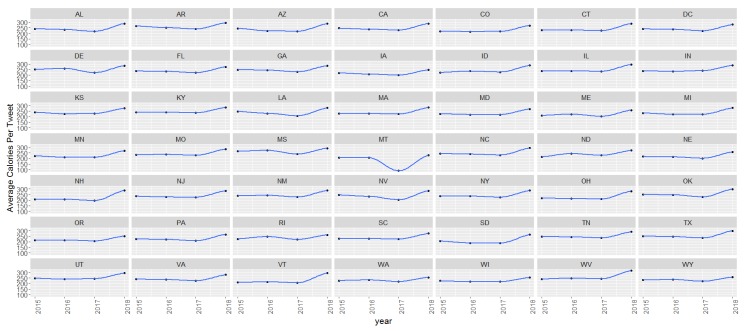
Temporal trends in average calories per tweet by state, Twitter 2015–2018.

**Table 1 ijerph-16-00975-t001:** Census tract level food environment characteristics and health outcomes (median regression).

Census Tract Characteristics	Prevalence of Obesity			Prevalence of Diabetes			Prevalence of Hypertension		
	β	SE	*p*-Value	β	SE	*p*-Value	β	SE	*p*-Value
N = 18,504 ^a^									
standardized mean calories ^b^	0.19	0.05	<0.001 *	0.02	0.02	0.192	0.16	0.04	<0.001 *
standardized % healthy food ^b^	−0.30	0.11	<0.001 *	−0.02	0.02	0.301	−0.13	0.04	0.001 *
standardized % fast food ^b^	0.15	0.04	<0.001 *	−0.02	0.01	0.055	0.12	0.03	<0.001 *

^a^ Census tracts with more than 10 tweets collected are included. ^b^ Twitter-derived food environment characteristics were standardized to have a mean of 0 and a standard deviation of 1. Adjusted median regression models were run for each outcome separately. Models controlled for census tract level demographics including population density, percent of 65 or older, percent of male, percent black, percent Hispanic, urban or rural areas, percent relatives besides spouse and children living in households, percent unmarried cohabitating adults, household size, percent owner-occupied housing, and income inequality. Data Source: American Community Survey, CDC 500 cities.

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
