# Peer review of "Census Tract Food Tweets and Chronic Disease Outcomes in the U.S., 2015–2018"

_ijerph, 2019, doi:10.3390/ijerph16060975_

Round 1

Reviewer 1 Report

Fascinating concept to analyze! I love the novelty of the study.

Some minor issues I caught: 

L47 - I think there is an inappropriately placed word (and) after the #13 reference

L70 - United States missing an "s" at the end

L83 - why only fast food restaurants? Was there data related to sit-down restaurants, convenience food items, etc.? Were those excluded or categorized differently? I don't think this is a huge methodological issue. I just think you need to justify why your focus was on the fast food component as compared to other food types, particularly because you were categorizing the tweets for this analysis. 

I think overall your methods are fairly solid approaches. 

Figure 4 - I am unable to determine what the x- and y-axis signify, thus making me unable to fully determine what you are showing here.

L137 - What are GEE models? I assume it's General Estimating Equation. I don't believe you have used this acronym previously, so you need to explicitly state what it is you are doing here

L176 - I think you also must address the limitation of those using Twitter. Does that represent your population? What is the average Twitter user? Why limit only to this social media? Would other geo-tagged posts from other social media sites have provided a more robust finding? I just think it was must be stated.  

Author Response

1.      L47 - I think there is an inappropriately placed word (and) after the #13 reference

Thank you so much. We’ve deleted the typo.

2.      L70 - United States missing an "s" at the end

Thank you for the catch! We’ve corrected this mistake.

3.      L83 - why only fast food restaurants? Was there data related to sit-down restaurants, convenience food items, etc.? Were those excluded or categorized differently? I don't think this is a huge methodological issue. I just think you need to justify why your focus was on the fast food component as compared to other food types, particularly because you were categorizing the tweets for this analysis. I think overall your methods are fairly solid approaches. 

Thank you for this comment and for your support of this manuscript. In the revised methods section, we add text describing our rationale for selecting fast food restaurants in particular.

Page 2, paragraph 4: “We chose fast food restaurants as an indicator of worse food environment because previous research has found that people who tweeted about fast food had more fast food restaurant in their surroundings than green retailers [18]. Living farther away from fast food restaurant has also been found to be associated with lower BMI for children [19].”

4.      Figure 4 - I am unable to determine what the x- and y-axis signify, thus making me unable to fully determine what you are showing here.

In the revised manuscript, we enlarge the font for the x- and y-axis. In addition, we now describe the axes in the manuscript text.

Page 6, Paragraph 1: “Figure 4 displays temporal trends for average calories by state. The x-axis displays year (2015-2018) and the y-axis displays average calories. The majority of states had an upward trend for average calories between 2015 and 2018.”

5.      L137 - What are GEE models? I assume it's General Estimating Equation. I don't believe you have used this acronym previously, so you need to explicitly state what it is you are doing here

We have replaced that sentence in the methods and now have written out the acronym.

Page 3, Paragraph 3: We additionally performed a generalized linear model using generalized estimating equations (GEE) to assess temporal trends in average calories per tweet, by state.

6.      L176 - I think you also must address the limitation of those using Twitter. Does that represent your population? What is the average Twitter user? Why limit only to this social media? Would other geo-tagged posts from other social media sites have provided a more robust finding? I just think it was must be stated.  

In the revised discussion section, we add text on our rationale for using Twitter as well as the study limitations of using Twitter.

Page 7, paragraph 3: “We utilized 1% publicly available tweets with geotags collected from Twitter’s free API service. It is estimated that about 28% of the online users possess a Twitter account in 2018 [28] and Twitter users tend to be younger in age [29]. Therefore, the information collected from Twitter may not be representative of all views expressed online or in-person at the neighborhood level. Facebook remains to be the most popular social media in the U.S. [29], however, the majority of profiles on Facebook are private whereas the majority of profiles on Twitter are public and thus available for analyses.”

Reviewer 2 Report

The changes in behavior, including nutritional behaviors of various population groups are influenced by internal and external factors. The Internet and social media are becoming one of the most important sources of information, including information on food and health. The assessment of information exchanged via the Internet and its impact on actual health behaviors seem to be an essential element of planning health education for specific population groups.

In order to better refer the results of this work to the real impact of the Internet on the occurrence of diet-related diseases, one should mention whether there was a factor linking directly to tweetting about food with obesity, diabetes and hypertension, for example, whether the 500 Cities CDC project included questions about the use of the Internet and social media.

It would also be interesting to assess the impact of gender and age, based on a study of 500 CDC cities and the census tract, on the observed relationship between the Twitter food environment and the occurrence of chronic diseases.

Author Response

1.      The changes in behavior, including nutritional behaviors of various population groups are influenced by internal and external factors. The Internet and social media are becoming one of the most important sources of information, including information on food and health. The assessment of information exchanged via the Internet and its impact on actual health behaviors seem to be an essential element of planning health education for specific population groups.

Thank you so much for your support of this manuscript. We do believe that harnessing information exchanged online can be a cost-effective way to understanding local health trends and providing information necessary for localized interventions.

2.      In order to better refer the results of this work to the real impact of the Internet on the occurrence of diet-related diseases, one should mention whether there was a factor linking directly to tweetting about food with obesity, diabetes and hypertension, for example, whether the 500 Cities CDC project included questions about the use of the Internet and social media.

The 500 Cities CDC project unfortunately does not specifically include variables about Internet or social media use. However, in our analysis, we account for rurality, minority composition and socioeconomic status—which correlate with internet access and use [1]. We add text to specify this as a potential limitation of the study (page 7, paragraph 3).

3.      It would also be interesting to assess the impact of gender and age, based on a study of 500 CDC cities and the census tract, on the observed relationship between the Twitter food environment and the occurrence of chronic diseases.

While the 500 Cities data did not have subgroup specific health outcomes, we were able to identify an alternative dataset with sex-specific obesity rates. We found that there were not statistically significant differences in the strength of associations between Twitter-derived variables and obesity rate for men vs. women.

Page 3, Paragraph 3: “Moreover, to assess the impact of Twitter-derived food environment on chronic outcomes in subgroups, we obtained county-level sex-specific obesity rates from CDC’s 2013 Behavioral Risk Factors and Surveillance System (BRFSS) – a telephone survey on health behaviors related to chronic diseases, injury, and preventable infectious diseases for the non-institutionalized, adult U.S. population. Age-adjusted estimates for percent obese by sex were generated from these data using a small area estimation technique [36,37]. Separate models were fit by sex to examine the association between Twitter-derived food environment characteristics and sex-specific obesity rates.”

Page 6, Paragraph 1: In analyses with sex-specific obesity rates, a one standard deviation increase in calories was associated with 0.32% increase in the prevalence of obesity in males and 0.42% increase in the prevalence of obesity in females. Similarly, a one standard deviation increase in percent fast food mentions was associated with 0.13% increase in the prevalence of obesity in males and 0.21% increase in the prevalence of obesity in females. Nonetheless, when we conducted a Chow test to assess the differences between the separate regression models, we found that there was no statistically significant difference between coefficient estimates for Twitter-derived variables in the male-specific obesity model versus those in the female-specific obesity model, suggesting similar strength of associations between Twitter food environment variables and sex-specific obesity rates.
